# Brown and Beige Adipose Tissue: One or Different Targets for Treatment of Obesity and Obesity-Related Metabolic Disorders?

**DOI:** 10.3390/ijms252413295

**Published:** 2024-12-11

**Authors:** Yulia A. Kononova, Taisiia P. Tuchina, Alina Yu. Babenko

**Affiliations:** 1World-Class Scientific Center “Center for Personalized Medicine”, Almazov National Medical Research Centre, 197341 St. Petersburg, Russia; yukonon@mail.ru; 2Endocrinology Department, Almazov National Medical Research Centre, 197341 St. Petersburg, Russia; tayka_91@mail.ru

**Keywords:** brown adipose tissue, beige adipose tissue, obesity, metabolic health, genetic pathways of browning, thermogenesis-affecting drugs

## Abstract

The failure of the fight against obesity makes us turn to new goals in its treatment. Now, brown adipose tissue has attracted attention as a promising target for the treatment of obesity and associated metabolic disorders such as insulin resistance, dyslipidemia, and glucose tolerance disorders. Meanwhile, the expansion of our knowledge has led to awareness about two rather different subtypes: classic brown and beige (inducible brown) adipose tissue. These subtypes have different origin, differences in the expression of individual genes but also a lot in common. Both tissues are thermogenic, which means that, by increasing energy consumption, they can improve their balance with excess intake. Both tissues are activated in response to specific inducers (cold, beta-adrenergic receptor activation, certain food and drugs), but beige adipose tissue transdifferentiates back into white adipose tissue after the cessation of inducing action, while classic brown adipose tissue persists, but its activity decreases. In this review, we attempted to understand whether there are differences in the effects of different groups of thermogenesis-affecting drugs on these tissues. The analysis showed that this area of research is rather sparse and requires close attention in further studies.

## 1. Introduction

Adipose tissue (AT) is a complex multifunctional organ that plays a fundamental role in the control of whole body energy homeostasis, insulin sensitivity (IS), blood pressure (BP), angiogenesis, inflammation and immunity by secreting various hormones and adipokines. In humans, AT is represented by several phenotypes. White adipose tissue (WAT) is specialized for energy storage. In contrast, the classic brown AT (cBAT) increases its expenditure and is responsible for thermogenesis. Beige AT or recruitable brown AT (rBAT) is a hybrid form of AT that shares characteristics with WAT and cBAT. It also contributes to energy consumption and heat production, but it is only caused by some factors such as physical activity, low temperatures and exposure to thermal substances of various origin [1].

To avoid confusion, the following terminology will be used in the current review when discussing human studies: cBAT is AT predominantly located in the adult in the supraclavicular, cervical, parasternal, and paravertebral regions. rBAT, which, in addition to the cervical and supraclavicular region, forms in the subcutaneous or visceral WAT by browning, will be referred to as rBAT. The term “browning” refers to the adaptive response of WAT to inducing factors, leading to the formation of beige and/or brown adipocytes.

Currently, there are several theories on the formation of beige adipocytes. These cells can differentiate de novo from various precursors (expressing PDGFR alpfa, CD34, SCA1, *MYH11*), as well as through the transdifferentiation of white differentiated (mature) adipocytes under different stimuli (Figure 1, Table 1). Additionally, there is the possibility of activating “sleeping” beige adipocytes. When both types are described generically, the term BAT will be used. This variant will be the most frequent, as it is only possible to clearly distinguish cBAT from rBAT by assessing the expression of genes specific for these subtypes (Table 1), and studies with their evaluation in the process of exposure to various factors are extremely rare in the literature [2]. This review will focus on attempting to distinguish between drug interventions that provide the activation of cBAT or rBAT.

## 2. Types and Value of Brown Adipose Tissue in a Human Body

### 2.1. The Role of BAT in Counteracting Obesity and Metabolic Syndrome

At the same time, beige adipose cells, as well as brown adipocytes, are a source of powerful biological factors (batokines) that play a significant role in the maintenance of metabolic health (MH) [3,4]. In the development of obesity, a number of changes occur in the AT, which are accompanied with its inflammation and fibrosis, reduction in its plasticity and depositing capacity. Also, obesity causes damage to the cytoarchitecture of the BAT, which leads to the whitening (de-browning) of brown adipocytes and impaired thermogenesis. At the same time, in experimental studies, de-browning was noted to be one of the earliest changes occurring in AT in obesity, and the restoration of normal browning status can significantly slow down further pathological remodeling of AT [5]. BAT dysfunction is characterized by a decrease in the level of markers of angiogenesis, an increase in the level of inflammatory markers and the development of oxidative stress.

### 2.2. The Role of BAT as a Therapeutic Target

In turn, the activation of BAT may improve metabolic parameters associated with obesity such as insulin resistance, dysglycemia and dyslipidemia [6]. The BAT is an important site of glucose utilization and triglyceride (TG) clearance, and it is considered a potential target in the fight against obesity and comorbidities [6].

Increased amounts of BAT are associated with better MH, lower body mass (BM), and smaller amounts of WAT, including visceral AT (VAT) [7,8]. When discussing the role of BAT as a therapeutic target, we should take into account that, although cBAT is less active in obese patients, its amount remains quite stable in adults. At the same time, the number of beige adipocytes, which are formed under the influence of thermogenic factors, (cold, physical activity, hormones, nutrients and drugs) can vary to a large extent and thus represent a more interesting target for obesity therapy or, more precisely, for preserving/restoring MH in obesity. The beige adipocytes can provide energy consumption due to the dissociation of oxidizing phosphorylation with uncoupling protein 1 (UCP1) and through independent from the UCP1 mechanisms (Table 1). One of them is the ATP-dependent calcium cycle [9], through which UCP1-independent thermogenesis is realized via sarco/endoplasmic reticulum Ca^2+^-ATPase 2b (SERCA2b) and ryanodine receptor 2. This significantly increases glucose consumption through enhanced glycolysis, the metabolism of tricarboxylic acids, and pyruvate dehydrogenase activity to provide ATP-dependent thermogenesis through the SERCA2b pathway. This suggests that rBAT may play a more important role in maintaining normal glucose metabolism than cBAT or the creatine cycle.

### 2.3. BAT Activity Estimation

In earlier studies, direct and indirect calorimetry was used to estimate BAT activity. Since the beginning of the 21st century, positron emission tomography/computed tomography (PET/CT) with 18F-fluoro-deoxyglucose (18F-FDG) began to be used as the reference method for estimating BAT activity. Meanwhile, both PET/CT and MRI show the presence of thermogenic tissues (BAT) but do not distinguish BAT from beige AT. Histologic, immunohistochemical, and molecular genetic studies of AT have shown that both cBAT and rBAT cells can be identified in different sites. In addition, as mentioned above, some of the rBAT cells are in a “dormant” state. The methods evaluating the metabolic activity of the tissue do not allow us to recognize all these subtle features [10,11]. In adults, cBAT accumulates in the cervical, supraclavicular (80%), paravertebral and parasternal regions, although biopsies and gene expression assessments have found both brown and beige adipocytes to be present in these areas [12].

To increase the metabolic activity of BAT, PET/CT with 18-FDG is performed using a cold protocol. However, with this approach, PET/CT may confuse us about the amount of BAT, for several reasons. First, only less than half of BAT depots is activated by cold exposure. When stimulated by physical activity, lower BAT glucose uptake has been reported in humans [13,14], which may be explained by a higher requirement of skeletal muscle for glucose uptake during periods of high physical activity, and, therefore, there is a decrease in its uptake by other tissues, including BAT. Second, animal studies have shown that different rBAT markers are expressed very differently under different temperature regimes. Whereas the *Tmem26* expression is lower under isothermal conditions and increases significantly under cooling, *Epsti1* expression is independent of inducers, while CD137 and TBX1 expression become the highest under isothermal conditions. At the same time, the marker of classical BAT, the *Zic1* gene, is not expressed in abdominal SAT, and the universal markers of brown and beige AT, *Ucp1* and *Lhx8*, have very scant expression in abdominal SAT, but, like the *Zic1* gene, are actively expressed in AT at sites of BAT localization [12]. Translating these data to humans is a matter of future research, but these data make us think that the informativeness of PET/CT is insufficient to assess the subtle mechanisms of regulation of BAT activity. The presence and activity of BAT can be assessed more precisely by the expression of universal marker genes of thermogenesis; for example, *UCP1* reflects (mediates) the process of dissociation of oxidative phosphorylation at the inner mitochondrial membrane (non-shivering thermogenesis). In addition, it can be evaluated by the secretion of BAT biological products—batokines (fibroblast growth factor 21 (FGF21), microRNA, Irisin/FNDC5 (fibronectin type III domain-containing 5), interleukin-6 (IL-6) and 12,13-dihydroxy-9 Z-octadecenoic acid). The evaluation of molecular genetic markers in AT is a specific method to evaluate the specific phenotype of BAT (brown or beige), and the activation of certain molecular genetic pathways of browning is the evaluation of molecular genetic markers involved in various processes (adipogenesis, browning, inflammation) in different adipose tissues. Browning is assessed by determining both the expression of brown (*UCP-1*, *PRDM16*, *PPARG*, *PPARGC1A*) and beige (*TMEM26*, *EPSTI1*, CD137, *TBX1*, *F2RX5*, *ZFP423*) AT marker genes in SAT and/or VAT, as well as the expression of microRNAs determining the inhibition or activation of browning or increased BAT activity (Table 1). Of particular interest is the evaluation of microRNAs with multidirectional activity. For example, miR378 inhibits browning in SAT but increases the activity of cBAT [15].

### 2.4. Factors Influencing on BAT Activation

The accumulated studies to date suggest that different activation mechanisms differ in their ultimate effect. We can distinguish the following ways of influence on BAT activation: (1). Physiologic exogenous stimuli: cold, stress, physical load. (2). Endogenous stimuli: increase in the level of catecholamines (CA), thyroid hormones (THs), batokines (sodium-uretic peptides, fibroblast growth factor 21 (FGF) 21, FGF19, prostaglandin E2, nitric oxide, adenosine, etc.). Many of the above stimuli can also act as exogenous drug effects: thyroid hormone preparations and adrenaline, prostaglandins, adenosine. Based on FGF21, a number of drugs have been developed. When discussing the drug effects, it is also worth mentioning nutritional components, but we will not dwell on them in detail as they are described in a recent review [16]. An exception will be made only for resveratrol, which has the most significant history of study and illustrates a mechanism of action via sirtuins (SIRTs). Finally, drugs that have been overwhelmingly developed for other purposes but have shown effects on BAT activity: drugs for the treatment of obesity and/or type 2 diabetes mellitus (T2DM): metformin, thiazolidinediones (TZDs), glucagon-like peptide (GLP)-1 receptor agonists (GLP-1RA), sibutramine; drugs for the treatment of overactive bladder: beta3-adrenoreceptor (AR) agonists (beta3-ARa) (mirabegron). To avoid repetition in describing effects with the same underlying mechanism, e.g., endogenous and exogenous excess of THs, CA, these effects will be described together.

As noted above, in humans, cold (cold-induced thermogenesis) or stress (beta3-adrenergic activation under conditions of sympathetic nervous system (SNS) hyperactivity) can act as inducers of browning. The best effect is produced by general cooling of the body. This approach is used in PET/CT protocols for the detection of active BAT: the patient is cooled (in light clothing at 16 °C) for two hours before the test is performed [17]. Meanwhile, obese patients have low adherence to cooling recommendations. The studies have shown the effective induction of browning with localized cooling as well. For example, a study conducted by Finlin et al. used a 30 min application of an ice block every day for 10 days to the upper thigh. Cold caused a significant increase in the expression of *UCP1*, *STAT3*, and *TMEM26* in both healthy subjects and obese patients, regardless of the age of the subjects. AT biopsies were performed before and after 10 days of cooling, both from the thigh exposed to cooling and on the contralateral side. There were unexpected correlations of *TMEM26* expression: positive correlation with body mass index (BMI) and a negative correlation with IS [18].

The presence of such relationships may be explained by the fact that, when exposed to cold, systemic adaptation mechanisms are triggered, which include both the enhancement of thermogenesis and free heat release (browning activation) and other mechanisms of body temperature maintenance—the enhancement of energy intake into the body (the stimulation of hunger via central sympathetic mechanisms) and the activation of the adipogenesis of SAT as a way to preserve heat by increasing the thickness of the subcutaneous fat layer. On the other hand, positive metabolic effects have been reported in longer studies with exposure to low temperatures. Thus, cold activation of the BAT for 30 days with a total cold exposure of 180 h was accompanied by an increase in IS [19], and the study for 42 days with a total cold exposure of 84 h (2 h per day) resulted in a fat mass reduction of 5.2% [20].

Meanwhile, even with respect to a single 2 h cold exposure as part of the 18F-FDG PET/CT study, we observed very low adherence—about a quarter of patients who performed this study at the first visit refused to repeat it at the follow-up examination. It is hardly realistic to expect high adherence to long-term interventions of this kind. Physical activity is a well-established method for improving metabolic health. Current data suggest an increase in IS and glucose uptake in the liver, AT and muscle with exercise. In addition, fatty acid (FA) oxidation and gluconeogenesis increase in the liver, and de novo lipogenesis and inflammation decrease. In muscles, in addition to an increase in their mass and strength, there is an increase in mitochondrial biogenesis and lipid oxidation, increased oxygen consumption by cardiomyocytes, the improved functional status of vascular smooth muscle cells and endothelial function, resulting in a decrease in heart rate and resting BP. In AT, fat deposition declines, and lipolysis rises. Even in the composition of the intestinal microbiome under the influence of physical activity, positive changes occur: alpha and beta diversity improves, and dysbiosis lowers [21,22]. Studies evaluating the effect of exercise on browning and BAT activity have not demonstrated convincing evidence of its activation. On the one hand, physical activity elevates a number of batokines: interleukin-6 (IL-6), β-aminoisobutyric acid, FGF21, sodium-uretic peptides, and circulating lactate. Experimental studies have shown that the effects of physical activity on BAT activity are mediated by the PGC1 alpha [23], the expression of which is stimulated by the release of the batokine irisin during exercise [24]. However, in a human study conducted by Vosselman et al., not only was there no increase in BAT activity, but, instead, a decrease was demonstrated during training [25]. Similarly, in a study conducted by Nakhuda et al., no significant change in the expression of the marker genes for BAT (*UCP1*, *PRDM16*, *LHX8*) and cBAT (*EVA1*) was detected from the evaluation of biopsies of AT localized in the perimuscular region [26].

In this regard, the issue of developing alternative methods of browning activation is extremely relevant. Active search is being conducted to identify the ability of drugs to increase browning, but the pronounced effects demonstrated in animal experiments for many drugs have been very moderate in clinical conditions. This is due to the fact that, in many cases, the systemic administration of drugs that activate browning requires significantly higher doses than the average therapeutic ones. The second problem is the development of significant side effects when using drugs at doses that activate browning when administered systemically (dinitrophenol, triiodothyronine). To overcome these problems, in recent years, methods of targeted drug delivery directly to the AT have been increasingly studied to provide effects on browning locally [27,28].

A well-studied effect on BAT activity is the increase in TH levels.

## 3. Inducers and Drugs Based on Them

### 3.1. Thyroid Hormones

#### 3.1.1. Basic Mechanism

The main mechanism is the conversion of thyroxine (T4) to triiodothyronine (T3) with the participation of type 2 deiodinase (DIO2), the gene for which is expressed in the cells of the BAT and is its marker. In addition, TH receptors (TRs) are expressed in the BAT and TH together with the noradrenaline increase of UCP1 expression through the effect of T3 on the TR beta, activating β3-AR through it. The SNS acts synergistically with THs to modulate BAT activity by increasing the expression of DIO2, which in turn increases the availability of THs in the BAT [29]. Conversely, T4 binding to TRs enhances the effects of beta3-AR stimulation on the activity of BAT [30]. Although THs can induce BAT browning even in the absence of sympathetic activation, their thermogenic potential will be significantly attenuated in this case [31]. SNS-independent direct effects of T3 on browning include direct effects on mitochondrial autophagy through the activation of SIRT1 and the inhibition of the activity of mTOR [32] (Appendix A).

#### 3.1.2. Experimental Studies

In a study on human subcutaneous preadipocyte cell lines, co-incubation with T3 resulted in a significant increase in *UCP1* expression and a substantial decrease in peroxisome proliferator-activated receptor (PPAR) gamma [33]. In experimental models, THs directly activate BAT by increasing brown adipogenesis [34] and may stimulate BAT indirectly through the increased production of thyroid-stimulating hormones [35]. In recent studies, long-term T3 treatment has been demonstrated to increase the recruitment of thermogenic capacity in the interscapular AT of male mice by TR alpha. In the study, mediated hyperplasia, promoting the proliferation of progenitor cells and consequently increasing the population of adipocyte precursors in the depot of rBAT. There was a decrease rather than an increase in UCP1 mRNA levels after long-term T3 treatment, although UCP1 protein levels per mg of tissue protein were elevated after both short- and long-term treatment with T3 [36,37]. In brown adipocytes, THs play an essential role in thermogenesis via UCP1 by acting at the level of the UCP1 enhancer, where TRs form a heterodimeric complex with the retinoic acid receptor and bind to the PGC1 alpha. TRs additionally interacts with cyclic adenosine monophosphate (cAMP) response element binding protein (CREB), acting synergistically with noradrenaline (NA) and increasing the expression of *UCP1* [38].

#### 3.1.3. Clinical Trials

Increased levels of both exogenous and endogenous THs can induce BAT. Patients with hyperthyroidism demonstrate a three-fold increase in glucose uptake in BAT measured by PET/CT with 18F-FDG compared to healthy subjects [39]. Broeders et al. studied the effect of levothyroxine at a mean dose of 137.75 ± 23.75 µg/day versus cold exposure on BAT activity measured by PET/CT in patients with highly differentiated thyroid carcinoma. The first study was performed before thyroxine therapy 6–8 weeks after surgical removal of the thyroid gland (hypothyroid state), and the second study was performed after 6 months of levothyroxine therapy (subclinical hyperthyroid state). A significant increase in energy expenditure as measured by indirect calorimetry after treatment was demonstrated (basal metabolic rate: 3.8 ± 0.5 kJ/min vs. 4.4 ± 0.6 kJ/min, *p* = 0.012). Basal metabolism increased from 15 ± 10% to 25 ± 6% (*p* = 0.009). The mean metabolic activity of BAT increased in thyroxine therapy (standard uptake value 4.0 ± 2.9 vs. 2.4 ± 1.8, *p* = 0.039) [40]. The effect of elevated endogenous T4 levels was examined in a study conducted by Heinen et al. [41], which included 16 healthy men without obesity who were randomized to the administration of 400 µg of thyrotrophin-releasing hormones (TRHs) or 2 mL of saline solution. One to three weeks after TRH administration, glucose uptake by BAT as measured by PET/CT was markedly increased in four and nine subjects, compared with the placebo. Analysis of the relationship between T4 concentrations and mRNA expression levels of browning markers in SAT and VAT samples obtained from a large cohort of patients showed that mRNA levels of *UCP1* and *CIDEA* in SAT and *PRDM16* in SAT and VAT correlated with circulating T4 levels [42]. Multifactorial regression analysis showed that free T4 levels in serum significantly contributed to changes in browning mRNA levels (*PRDM16*, *CIDEA* and *UCP1*) [43]. In a study conducted on adipocytes obtained from postmortem biopsies of infant’s subscapular AT, co-incubation with T3 significantly increased mRNA levels of PGC1 alpha (2.5 times, *p* < 0.05), *UCP1* (10 times, *p* < 0.001) and *UCP1/FABP4* (10 times, *p* < 0.001). However, when BM loss was attempted by administering high-dose thyroxine, its loss was noted only early (first 8 weeks) [44], which correlates with the data of experimental studies. Moreover, the development of serious cardiovascular (CV) complications during prolonged exposure with high TH levels is a well-known fact [45,46] that excludes the use of TH administration for weight loss but leaves the task of their targeted delivery to the SAT to activate browning relevant. This option provides an increase in the volume and activity of both cBAT and rBAT.

### 3.2. Catecholamines

The basic mechanism: increased plasma CA levels are a major endocrine mechanism that, acting via cAMP and protein kinase A (PKA), phosphorylates adipose triglyceridlipase stimulating lipolysis [47]. By activating beta3-AR, they stimulate the p38MARC pathway via PKA, increasing the expression of *ATF2* and PGC1 alpha. As an endogenous increase in the stress, as well as the exogenous administration of NA and adrenaline, the increased energetic expenditure and thermogenic activity in the BAT depends on intact sympathetic stimulation [48]. Meanwhile, non-selective catecholaminergic hyperactivity, while having some pluses (an increase in basal metabolic rate and browning), is characterized by a huge number of minuses (tachycardia, arterial hypertension (AH), the development of CA-dependent cardiopathy and heart failure) observed, for example, in pheochromocytoma [49] (Appendix A).

Drugs that moderately increase catecholaminergic activity include neurotransmitter reuptake inhibitors, particularly sibutramine and beta-ARa.

### 3.3. Sibutramine

This drug inhibits the reuptake of neurotransmitters. By inhibiting serotonin reuptake, it reduces hunger, accelerates satiety, and contributes to reduced food consumption.

#### 3.3.1. Basic Mechanism

The inhibition of NA reuptake may provide a peripheral effect on thermogenesis and the activation of BAT by mediating the activation of beta3-ARs (Appendix A).

#### 3.3.2. Experimental Studies

In experimental studies, sibutramine demonstrated a pronounced enhancement of UCP1 and thermogenic mitochondrial protein expression in the BAT [50]. Browning activation in WAT was observed only in one-third of cases. Increasing the drug concentration in SAT using targeted delivery will possibly allow us to demonstrate an increase in browning activity, given the universality of the mechanism of beta3-AR activation for SAT and browning.

#### 3.3.3. Clinical Trials

In humans, the development of a thermogenic response has been demonstrated in both the single administration of sibutramine [51,52,53] and in the long-term (12 weeks) treatment of obese patients (thermogenesis increased from 1.27 ± 0.29 kcal/kg/h to 1.44 ± 0.13 kcal/kg/h on sibutramine and decreased from 1.56 ± 0.27 kcal/kg/h to 1.33 ± 0.36 kcal/kg/h on placebo). We studied the effect of sibutramine on the expression of some molecular genetic markers of browning in SAT in obesity [54] and did not observe a significant change in the expression of miR-378, involved in the increase in browning activity in cBAT and the inhibition of browning in WAT. *UCP1* expression was detected in only two patients out of thirty-two before treatment and in three after treatment, making it impossible to assess the effect of sibutramine on its expression. At the same time, both the maximum (SUV max (g/mL) 2.71 ± 2.52 before treatment and 3.74 ± 1.45 after treatment, *p* = 0.01) and mean (SUV mean (g/mL) 1.79 ± 1.57 before treatment and 1.98 ± 0.84 after treatment, *p* = 0.02) metabolic volume of BAT according to PET/CT data increased in therapy [8]. Meanwhile, in the oral administration, sibutramine has a number of limitations, determined just in the greatest degree by its catecholaminergic effects (sleep disturbance, dry mouth, increased BP and tachycardia). This limits the use of this drug in those patients who are in high need of browning activation—patients with CV disease who cannot significantly increase physical activity to increase basal metabolism. Overall, it seems that sibutramine can activate not only the cBAT, but also the browning, but increasing the activity of the latter requires either a higher dose, which is limited by side effects or local delivery to the AT.

Based on the fact that beta3-ARs are mainly involved in the activation of AT browning, selective drugs will be safer in drug therapy aimed at browning activation. Accordingly, the study of the effect of beta3-ARa on BAT has received the greatest development.

### 3.4. Beta3-Adrenoreceptor Agonists

Mirabegron is currently registered for clinical use, but the indication for use is overactive bladder.

#### 3.4.1. The Basic Mechanism

The basic mechanism is the activation of beta3-AR via the p38-MAPK signaling pathway. The activation of different types of β-AR is involved in the regulation of BAT. The agonism of beta 1-ARs mainly stimulates BAT recruitment, and beta3-ARs mainly activate the thermogenic mechanism in mature brown adipocytes [55,56]. The binding of CA to receptors on the plasma membrane of the brown adipocyte leads to an increase in intracellular cAMP and the subsequent activation of cAMP-dependent PKA, which phosphorylates target proteins and genes responsible for the dissociation of mitochondrial respiration. CA binding also mediates fat oxidation and the release of free FAs, which contribute to *UCP1* activation [55,56] (Appendix A).

#### 3.4.2. Experimental Studies

In an in vitro study, Cao et al. showed that beta3-ARa stimulate the browning of WAT by increasing *UCP1* expression [57]. The authors demonstrated that the effects of beta3 stimulation on the browning of WAT are established through the p38-MAPK signaling pathway, and the addition of a p38-MAPK inhibitor abrogated these effects. The administration of a beta3-ARa provided pleiotropic effects, including increased lipolysis and improved IS. Mirabegron has been studied as a drug for the activation of BAT in rodents and humans. In obese mice, the administration of mirabegron resulted in 12% lower BM and a 14-fold increase in *UCP1* expression compared to controls [55]. In in vitro experiments, mirabegron stimulated increased *UCP1* expression and browning in 3T3-L1 white preadipocytes and in brown preadipocytes of mice [57].

#### 3.4.3. Clinical Trials

Cypess et al., when administering mirabegron to healthy men for 12 weeks, noted higher BAT activity and metabolic rates at rest compared to the group taking placebo [56]. Mirabegron significantly increased BAT activity as measured by PET/CT with 18F-FDG in all participants (*p* = 0.001). The authors calculated that the observed increase in energy expenditure could provide a weight loss of 5 kg/year. However, the dose selected for the study (200 mg/day) was higher than the dose approved by the FDA for clinical use (50 mg/day for the treatment of overactive bladder) and caused CV side effects such as increased BP and heart rate due to the activation of beta1- and beta2-AR. O’Mara et al. used a dose of 100 mg/day and found that 4-week treatment with mirabegron significantly increased BAT metabolic activity by 18F-FDG PET/CT, resting energy expenditure, high-density lipoprotein cholesterol and IS in healthy women, but was accompanied by CV symptoms and headaches in a proportion of participants [58]. We should note that none of these studies demonstrated significant weight loss, which may be explained by their short-term duration and raised the question of the need for longer studies. It should also be taken into account that only high doses of mirabegron demonstrated the activation of BAT in healthy subjects, while the FDA-approved dose of 50 mg/day did not significantly increase resting energy expenditure or result in weight loss in obese subjects with IR, despite the long duration of the intervention (12 weeks) but did induce SAT browning and improved beta-cell function [56]. In addition, this study demonstrated that mirabegron treatment reduced triglyceride levels in skeletal muscle and increased expression of the PGC1alpha (*p* < 0.05) and the number of type I fibers (*p* < 0.01). The PGC1alpha is part of a transcriptional network that regulates muscle fiber type determination and promotes FA oxidation, mitochondrial biogenesis, and type I fiber formation. Based on the fact that skeletal myocytes do not express β3-AR, these data indirectly suggest endocrine effects, probably through the batokines secreted by browning-prone SAT on muscle fiber differentiation.

Thus, it is of interest to study the effects of low and medium doses of mirabegron over longer periods of time (optimally 6 months) to evaluate the effects on BM loss. Meanwhile, the transdermal delivery of a β3-ARa (CL-316243) to the SAT using micro-needles demonstrated pronounced effects on browning at a much lower effective dose (5 mg daily) [29], herewith a significant BM loss and increased *UCP1* expression were observed [59], This allows us to expect that transdermal administration in SAT will provide high efficacy of β3-ARa at significantly lower doses, thus avoiding side effects and reducing treatment costs.

In general, the activation of those molecular genetic pathways involving THs, CA, and drugs modulating their activity (sibutramine and β3-adrenomimetics) provide increased representation and activity of both cBAT and rBAT. Meanwhile, all of the above agents are characterized by serious side effects on the CV system when administered systemically, especially when administered long-term and/or at high doses. This makes them poor candidates for improving MH, especially in those subpopulations of obese patients who need it most—aged patients, patients with CV disease and CV risk factors such as AH. Their further promotion as drugs for the activation of cBAT and browning should be considered mainly in the variant of targeted delivery to the AT.

The systemic effects of drugs that can be conditionally called sirtuin mimetics, such as metformin and resveratrol, differ significantly. Resveratrol is not an official drug but a dietary supplement, but, due to its high level of research in clinical trials, it will be discussed in this review. The previously described drugs indirectly affect SIRT1; however, the involvement of other molecular genetic pathways significantly modulates their total effect. Among the modulators of various pathways that are involved in the regulation of the browning process, sirtuins attract particular attention. Sirtuins are nicotinamide adenine dinucleotide class III histone deacetylases. Currently, seven different isoforms (SIRT1-7) have been described in mammals, and all of them, except for SIRT4, are involved in the activation of cBAT and rBAT. SIRT1 is one of the key modulators of the binding activity of PPAR gamma and, through its modulation (deacetylates), activates PGC1 alpha and PRDM16, induces typical BAT genes and inhibits WAT genes in the process of transdifferentiation in white adipocytes. SIRT1, through the activation of PGC1 alpha, promotes the transcription of SIRT3, which is crucial for mitochondrial biogenesis and metabolism. Given that SIRT1 and SIRT7 are more highly expressed in WAT, while SIRT3 and SIRT5 are mainly expressed in BAT, it can be assumed that SIRT1 has a greater influence on the formation of beige adipocytes in WAT, providing the process of transdifferentiation of white adipocytes into beige and inducing adaptive thermogenesis of WAT through the AMPK/SIRT1/PGC1 alpha pathway. Probably, both metformin and resveratrol increase the expression of SIRT1, but resveratrol acts on it directly, binding to the three-helix bundle of the N-terminal domain and, through it, activating AMPK. Metformin, on the contrary, activates AMPK and, through it, activates SIRT1. The additional enhancement of AMPK activity by metformin is provided by activating SIRT3. SIRT3 increases the expression of PGC1 alpha and *UCP1* in BAT directly—through the activation of ADP-ribosyltransferase and deacetylase, and indirectly through the phosphorylation of CREB, contributes to the transdifferentiation of white adipocytes into beige adipocytes, acting on *UCP1* through PGC1 alpha, *PRDM16* and *NRF1* under cold exposure [60,61]. SIRT5 has recently been found to be required for the differentiation and activation of brown adipocyte adipogenesis in vitro. The inhibition of SIRT5 leads to the impaired expression of CCAAT/enhancer binding protein beta in brown adipocytes, which in turn reduces the UCP-1 transcriptional pathway. *UCP1* transcription mediated by the SIRT5-C/EBP beta axis is possibly crucial for the regulation of energy balance and obesity-related metabolism [62]. In the experiment, SIRT5 knockout mice demonstrate a lower ability to darken in SAT and cold intolerance compared to the control group, indicating the significance of this sirtuin for thermogenesis in vivo [63].

### 3.5. Resveratrol

#### 3.5.1. Basic Mechanism

Resveratrol, a natural phenol and phytoalexin produced by several plants, is a known activator of the sirtuin family (especially SIRT1 and 3). Through the activation of SIRT1, it increases the expression of PPAR gamma and *PRDM16* [50] (Appendix A).

A significant disadvantage of resveratrol as a food supplement is its instability. Native resveratrol is sensitive to elevated temperature, pH and light due to the instability of its underlying molecular structure (C-C double bond and hydroxyl groups). The solution to this problem was the development of trans-resveratrol, which is stable in acidic environments and at room temperature.

#### 3.5.2. Experimental Studies

In experimental studies, resveratrol reduced AT inflammation by decreasing the expression of pro-inflammatory cytokine tumor necrosis factor alpha (TNFalpha) and IL-6 in the AT. Simultaneously, the increased expression of genes related to AT browning, including *Ucp1* and *Ppargc1a*, *Sirt1* and AMPK, and improved mitochondrial function were observed [50,64,65]. Trans-resveratrol also induced the upregulation of *Ucp1* and other markers of browning in the experiment [66].

#### 3.5.3. Clinical Trials

In clinical trials in first-degree relatives of T2DM patients in whom glucose uptake was assessed by PET/CT with 18F-FDG, cBAT activity did not change significantly on resveratrol therapy [66]. A study conducted by Bosveik et al. examined glucose uptake by PET/CT with 18F-FDG in SAT and VAT in patients with T2DM. The authors noted that, in SAT, 18F-FDG uptake did not change significantly after treatment with resveratrol (TBRMax 1.7 (1.6–1.7)) vs. placebo (1.5 (1.4–1.6); *p* = 0.05). But, in VAT, the uptake of 18F-FDG was increased (*p* = 0.024) [3]. In addition, resveratrol improved glycemic and lipid profiles in this study. In single clinical trials, the effects of resveratrol on molecular genetic markers of BAT in humans have been studied [67], including the decreased expression of pro-inflammatory cytokines (TNFalpha and IL-6) and the increased expression of thermogenesis markers (*UCP1*, *PRDM16*, PGC1 alpha). Twenty male and female volunteers aged 30–55 years, BMI ≥ 30 kg/m^2^ were divided into two groups, receiving 500 mg of trans-resveratrol or placebo for four weeks. SAT biopsies were taken before and at the end of the treatment period. Resveratrol was shown to improve glycemic and lipid profiles along with increased levels of UCP1, PRDM16, PGC1 alpha and SIRT1. In recent studies, the tissue delivery of resveratrol has also attracted attention as an opportunity to significantly enhance its effects on AT. Zu et al. synthesized liposomes (R-lipo) and encapsulated lipid nanoparticles (R-nano) containing trans-resveratrol. R-lipo had higher stability than R-nano, while R-nano had a longer release period than R-lipo. Both delivery methods increased resveratrol content in 3T3-L1 cells. Both R-nano and R-lipo dose-dependently induced the expression of UCP1 mRNA, the rBAT marker CD137, and suppressed the expression of the WAT marker IGFBP-3, to a greater extent than oral administration. R-lipo significantly induced the expression of UCP1 mRNA, while stimulation with isoproterenol enhanced the effect of R-lipo on PGC1 alpha expression. The expression of another rBAT marker, Tmem26, decreased under basal conditions but was increased under the influence of all forms of resveratrol following stimulation with isoproterenol [65]. Thus, for resveratrol, we found no convincing evidence in the literature for the increased expression of specific markers of cBAT, but there is evidence for the increased expression of rBAT markers and universal thermogenic markers.

### 3.6. Metformin

#### 3.6.1. Basic Mechanisms

In the study, on VAT adipocyte and hepatocyte cell lines, metformin significantly increased the expression of genetic markers of BAT (*UCP1*, *PGC1a*, *PRDM16*, *Elovl3* and *CIDEA*) through SIRT3 and 1a AMRK [68]. As an additional, indirect mechanism of metformin, we can consider an increase in lactate production, which is also an inducer of browning [3,69]. The detailed role of lactate elevation in the implementation of the positive metabolic effects of metformin was discussed in our earlier review [70]. Additionally, metformin is known as a powerful modulator of the microbiota, leading to a significant increase in the production of short-chain FAs [71] (Appendix A).

#### 3.6.2. Experimental Studies

In an experiment with C57Bl/6 mice, metformin increased thermogenic markers in BAT (UCP1, PGC1alpha) via adrenergic stimuli and FGF21 [50]. The effects of metformin on the expression of genes specific to brown and rBAT were evaluated in studies on 3T3-L1 cell lines and in VAT [48]. Metformin was found to increase the expression of universal thermogenic genes (*Ucp1*, *Cidea*, *Cox7a1*, *Ppargc1a*, *Prdm16*, *Elovl3*) [50]. In cells subjected to differentiation in the presence of metformin, an intensification of the expression of specific rBAT genes *Pat2* and *Tmem26* was observed [72].

#### 3.6.3. Clinical Trails

Clinical trials have attempted to evaluate the effects of metformin on cBAT activity and the production of some batokines. In a study conducted by Oliveira FR et al., women with polycystic ovary syndrome (PCOS) were randomized to receive metformin (1500 mg/day, *n* = 21) or placebo (*n* = 24) for 60 days. BAT activity was assessed by 18F-FDG PET/CT and plasma irisin levels. The groups were similar in terms of age, BMI, metabolic profile, and PCOS phenotypes. BAT activity was not significantly altered in women receiving metformin (median change in SUVMax = −0.06 g/mL, *p* = 0.484) compared to placebo. Plasma irisin levels also remained unchanged in the groups receiving metformin (median change = −98 ng/mL, *p* = 0.310) and placebo (median change = 28 ng/mL, *p* = 0.650) [73]. At the same time, we did not find clinical trials demonstrating the effects of metformin on molecular genetic markers of browning in AT in the available literature, although its extensive range of effects on metabolic parameters (weight [74], lipids and fibrinolysis [75,76]) allows us to hope for their presence. Indirectly, the effects of metformin on browning have been demonstrated by the modulation of FGF21 levels in patients with metabolic syndrome and HIV infection [77]. As pharmacokinetic studies have shown, the absence of pronounced effects of metformin on browning in humans can be explained by its low bioavailability in the AT in humans, which is only 40%. Thus, when metformin is administered orally at therapeutic doses of 1000–3000 mg/day, the blood concentration of metformin in humans is less than 40 mg. The dose per kilogram of BM in an obese patient is on average 20 mg/kg, and an overdose of metformin can lead to a complications [78,79]. Animal studies examining WAT browning have used dosages of 200–250 mg/kg/day, which is about 7–10 times the maximum daily dose for humans [80,81,82]. Moreover, the transdermal delivery of metformin using micro-needles induced the browning of WAT [24].

Another group with a different way of realizing effects from other drugs is the group of PPAR agonists, including PPAR gamma and PPAR alpha agonists.

### 3.7. Thiazolidinediones

#### 3.7.1. Basic Mechanisms

TZDs, PPAR gamma agonists, are widely used for the prevention and treatment of T2DM. TZDs enhance adipocyte browning through MAPK and PI3-K signaling pathways [83]. PPAR gamma perform critical and heterogeneous functions in AT, which include adipocyte differentiation and lipid deposition, with PPAR gamma being considered the main regulator of adipogenesis, and no other factors can induce adipogenesis without the presence of PPAR gamma. The PPAR gamma/PRDM16/EBF2/EHMT1 complex defines the identity of brown/beige adipocytes, and it is PPAR gamma that attracts PRDM16, EBF2 and EHMT1, ensuring the coordination of transcriptional circuits toward the brown/beige differentiation pathway [83,84]. To acquire brown/beige adipocyte identity, PPAR gamma recruits *PRDM16* to form a core transcriptional complex that determines brown adipocyte development from a skeletal muscle cell or beige adipocyte transdifferentiation from a white adipocyte. PPAR gamma attracts EBF2 and coactivates the expression of genes selective for BAT, such as *UCP1*, *PPARGC1A*, and *PRDM16* [85]. After the differentiation of brown/beige adipocytes, the PPARgamma/PRDM16 complex engages a different set of cofactors to ensure the functional activity of BAT/rBAT in adaptive thermogenesis and energy homeostasis, among which PGC1 alpha plays a central role. In several studies, PPAR gamma agonists have demonstrated the ability to induce brown adipocyte formation in the AT in vitro and in vivo through mechanisms involving PPAR gamma, C/EBPalpha PGC1 alpha-mediated pathways upon systemic administration. Importantly, PPAR gamma is a key modulator of adipogenesis in both white and BAT [83,84] (Appendix A).

#### 3.7.2. Experimental Studies

Experimental studies have unequivocally demonstrated the ability of TZDs to activate BAT. PPAR gamma expression is predominantly restricted to AT and is the main regulator of adipogenesis, promoting the differentiation of preadipocytes into adipocytes [86]. Selective activators of PPAR gamma, including TZDs, promote the transcription of brown fat genes in white adipocytes both in vitro and in vivo [86] through SIRT1, PRDM16, C/EBP alpha and PGC1 alpha-mediated mechanisms [86]. Mechanistically, these drugs act by direct binding and activating PPAR gamma and PPAR response elements (PPREs) on the promoter and/or enhancer of genes specific for cBAT. TZD-induced browning is not associated with increased energy expenditure or weight loss in vivo. However, treatment with rosiglitazone at a dose of 10 mg/kg for 10 days increased the expression of cBAT-specific genes (*Ucp1*, *Cidea* and *Cox8B*), in SAT, and, to a lesser extent, in VAT depots [87], the mRNA of *Prdm16*, a dominant regulator of cBAT development, was moderately increased by exposure to TZDs. This indicates that *Prdm16* is required for brown adipocyte development under TZD exposure. Under conditions of low levels of PRDM16 protein, PPAR gamma activation induced the differentiation of white adipocytes, whereas, under conditions of high levels of PRDM16 protein, PPAR gamma activation promoted the darkening of white adipocytes, PPAR gamma agonists induced the BAT gene program [88]. Given that TZDs have a number of significant side effects when administered systemically, including fluid retention and increased bone resorption in at-risk groups, attempts have also been made for these drugs to optimize their effects on browning by targeting delivery to the SAT. A mouse model of diet-induced obesity showed that, when rosiglitazone in nanoparticles was injected into the vasculature of the AT, there was an enhancement of both browning and angiogenesis in WAT [25]. Another team used rosiglitazone nanoparticles integrated into micro-needles as a browning-inducing agent. We should note that, as a drug that enhances adipocyte differentiation, rosiglitazone produces significant weight gain when administered systemically. When using micro-needle administration, a significant slowing of weight gain and improvement of cholesterol, TG and insulin was observed [89].

#### 3.7.3. Clinical Trials

The effect of pioglitazone therapy on glucose uptake as measured by PET/CT with 18F-FDG was studied on a group of 14 men without obesity, T2DM, or CV disease who were randomized to placebo (lactose; *n* = 7, age 22 ± 1 years) or pioglitazone (45 mg/day, *n* = 7, age 21 ± 1 years) for 28 days. Before treatment, subclavicular AT was taken for culturing and evaluating the effects of the drugs on molecular genetic markers of browning in vitro in human adipocytes. In this group, pioglitazone did not increase but rather decreased the uptake of 18F-FDG. At the same time, in an in vitro study, co-incubation with both pioglitazone and rosiglitazone showed an increase in UCP1 and β-AR expression in adipocytes [90]. Meanwhile, the disadvantage of this group of drugs is the activation of adipogenesis of both white and brown adipocytes, which leads to the fact that BM not only does not decrease, but, on the contrary, increases during therapy, despite a significant improvement in MH parameters.

### 3.8. PPAR Alpha-Agonists

#### 3.8.1. Basic Mechanism

The basic mechanism is that PGC1 alpha coactivates a number of nuclear hormone receptors, including not only PPAR gamma but also PPAR alpha, which are involved in the transcription of brown fat genes [91], in contrast to the activation of PPARbeta/delta [92,93]. In the liver, PPAR alpha induces the secretion of FGF21, a potent stimulator of BAT activity [94,95] (Appendix A).

#### 3.8.2. Experimental Studies

In a model of genetically determined obesity in mice, PPAR alpha activation caused browning through the induction of *UCP1* transcription. In fenofibrate-treated animals, there was an increase in PPAR alpha and PPAR beta expression, as well as an associated increase in PGC1 alpha, *Prdm16*, *Bmp8b* gene expression [84]. Animals fed a high-fat diet in combination with fenofibrate had significantly higher energy expenditure and 11% lower BM with comparable feed intake than animals on a high-fat diet without fenofibrate. They were also characterized by significantly lower glucose, insulin, leptin and higher irisin levels. They had a significantly lower adiposity index and adipocyte diameter. In another study, fenofibrate also provided the expression of thermogenic genes, including *Ppara*, *Ppargc1a*, *Tfam*, *Prdm16*, beta3-AR, *Bmp8b* and *Ucp1* in brown adipocytes, which allowed PPAR alpha to be considered as a therapeutic target for the correction of metabolic disorders through the induction of browning [91].

In cBAT in mice, PPAR alpha shares genomic binding sites with PPAR gamma (14,233 common binding sites). Meanwhile, PPAR alpha knockout in cBAT had no significant effect on PPAR gamma or UCP1 expression, BAT volume and activity. Thus, PPAR alpha activation is not necessary to enhance the activity of cBAT and browning, but dual PPAR alpha/gamma activators are better at activating white fat browning in vitro and in vivo than PPAR gamma alone. Dual PPAR alpha/gamma agonists induce WAT browning in vivo, in part through increased FGF21 production in the liver. The action of PPAR gamma in AT and PPAR alpha-mediated increase in FGF21 synergistically induce WAT browning in vivo [84].

### 3.9. Selective PPAR Gamma-Modulator, Imatinib

Imatinib, an established therapeutic agent for chronic myeloid leukemia, is an effective tyrosine kinase inhibitor with pleiotropic effects on PPAR gamma.

#### 3.9.1. Basic Mechanisms

The phosphorylation of PPAR gamma at SER273 (pS273) is associated with obesity and insulin resistance. This phosphorylation does not globally alter the transcriptional activity of PPAR gamma but rather dysregulates a specific set of genes involved in obesity and T2DM. Both TZDs and selective PPAR gamma modulators, particularly imatinib, inhibit CDK5-mediated PPAR gamma pS273, thereby blocking the phosphorylation of PPAR gamma that it mediates [96]. Imatinib significantly enhances the expression of thermogenic genes, including *UCP1*, *PPARGC1A*, *COX5B*, along with the expression of markers for beige adipocytes (*CD137* and *TMEM26*). Conversely, pro-inflammatory genes such as IL-6, monocyte chemoattractant protein-1 (MCP1) and TNF alpha were significantly reduced in WAT following treatment. The hyperexpression of PPAR gamma S273A promoted the IL-4-mediated activation of M2 macrophages.

#### 3.9.2. Experimental Studies

In mice fed a high-fat diet, imatinib improved IS and glucose levels without affecting body mass. Unlike TZDs, imatinib neither impacts weight nor induces fluid retention, nor does it affect hemodilution or the expression of genes involved in bone tissue formation, including bone sialoprotein, osteocalcin and osterix [96].

The drug is registered as an anti-cancer agent (for the treatment of chronic myeloid leukemia) and has been minimally studied in humans concerning its effects on BAT. Its widespread use for the improvement of MH is unlikely, given the relatively high incidence (approximately 30%) of severe side effects, which include leukopenia, anemia, thrombocytopenia, nausea, vomiting, fluid retention leading to edema, diarrhea, bleeding, fever, muscle cramps and bone pain.

### 3.10. Erythropoietin

#### 3.10.1. Basic Mechanisms

Erythropoietin (EPO) is a product of endocrine cells in the kidney that regulates erythropoiesis. Erythropoietin is registered as a hematopoietic drug, but its pleiotropic effects have been shown to affect BAT. EPO has demonstrated the ability to upregulate PRD1-BF1-RIZ1 homologous domain containing 16 (PRDM16), a transcription factor that plays a crucial role in brown adipocyte differentiation. This effect is realized through increased STAT3 activity, which stabilizes PRDM16 and the suppression of myocyte enhancer factor 2c (Mef2c) and microRNA-133a (miR-133a) via the β3-adrenergic receptor, which in turn increases UCP1. In addition, EPO increases the production of the batokine FGF21 in BAT. Another recent study demonstrated that EPO regulates energy metabolism in male mice via the EPO-EpoR-RUNX1 axis [97].

#### 3.10.2. Experimental Studies

The above described mechanisms were demonstrated in studies on C57BL/6J mice treated with a high fat diet and randomly (half of the group) with an intraperitoneal injection of recombinant human EPO (200 IU/kg) (HFD-EPO) three times a week for four weeks. The mice injected with EPO showed significantly lower body weight, epididymal and subcutaneous BAT mass, HOMA-IR and glucose levels by the end of the experiment, despite the same caloric intake and motor activity compared to the control group. A molecular genetic study demonstrated the above described changes [98,99]. Meanwhile, the results of experimental studies created the impression that, to consider EPO as a therapeutic target, genetic specificity should be considered, and, in general, its usefulness should be tested in humans. The realization of its effects was limited by the estrogen receptor (ER alpha), and its effects on BAT volume reduction and BAT activation were fully represented in males but in females only when ER alpha was deleted. We also found no data on the effect of EPO on beige adipocytes and their specific genes in the available literature. Experimental data demonstrated a decrease in SAT in mice, and, although in the experiment, it was associated with an improvement in metabolic parameters, the verification of this information in clinical trials is necessary [100].

Despite the large number of experimental studies on EPO, we found no clinical trials in the available literature on its effects on BAT, demonstrating the need for further research (Appendix A).

### 3.11. Angiotensin 1-7

#### 3.11.1. Basic Mechanism

The role of modulation of renin–angiotensin–aldosterone system (RAAS) activity in the maintenance of metabolic health is well known. However, its contribution to BAT activity is poorly understood. Angiotensin 1-7 (Ang1-7), a key protective peptide of the RAAS, is produced from angiotensin II (Ang II) and angiotensin 1-10 by the action of angiotensin-converting enzyme 2 and neutral endopeptidase, respectively. The specific receptor for Ang1-7, the Mas receptor is highly expressed in BAT.

#### 3.11.2. Experimental Studies

In experimental studies, the administration of angiotensin 1-7 (Ang1-7) has demonstrated the ability to activate BAT and thermogenesis and increase the expression of the β3-adrenergic receptor, leading to increasing norepinephrine signaling and, through this mechanism, increase the expression of *UCP1*, *PRDM16*. *PRDM16* expression was upregulated in concordance with AMPK upregulation and mTOR phosphorylation. AT weight, glucose and lipid levels were lower in Ang1-7-treated mice [101]. More recent studies demonstrated central (in central nervous system) effects of Ang1-7 on BAT activation [102]. Meanwhile, neither studies demonstrating the specific activation of beige adipocytes nor human studies confirming the effects of Ang1-7 in humans were found (Appendix A).

### 3.12. Nuclear Receptors and Ligands: Agonists of Intestinal Farnesoid X Receptor (FXR) (Fexaramine)

#### 3.12.1. Basic Mechanism

The action on FXR through the administration of Fexaramine (Fex) leads to a sustained increase in the production of FGF15, which activates the thermogenic program in BAT. Furthermore, Fex promotes the phosphorylation of p38, indicating the involvement of this gene in the mechanisms through which FXR agonists exert their effects. Gene expression analysis during Fex treatment demonstrated the induction of the gene coding for estrogen receptor gamma, which was associated with an increase in the expression of *PPARGC1A* (the gene encoding PGC1 alpha) and *PPARGC1B* (the gene encoding PGC1beta), as well as several of their target genes involved in thermogenesis, mitochondrial biogenesis, and FA oxidation in BAT.

#### 3.12.2. Experimental Studies

In DIO mice receiving Fex, there was a significant reduction in lipid levels, increased activity of protein kinase A, and the phosphorylation of p38, along with an elevation in body temperature, indicating a coordinated activation of thermogenesis in BAT. The number of adipocytes expressing UCP1 was significantly increased, as was the expression of PGC1 alpha, PRDM16 and PPAR gamma in animals treated with Fex [103].

#### 3.12.3. Clinical Trials

In the available literature, no studies have reported on FXR agonists evaluating the expression of thermogenic genes based on adipose tissue biopsy data. In randomized trials, the effects of another FXR agonist, obeticholic acid (OCA), were studied in a group of patients with T2DM and non-alcoholic fatty liver disease (NAFLD) (23 in the placebo group, 20 receiving 25 mg of OCA, and 21 receiving 50 mg of OCA) regarding IS, metabolic parameters and levels of the FGF19 [104]. FGF19, derived from the intestine, is a crucial regulator of bile acid and glucose metabolism; in humans, circulating levels of FGF19 are directly associated with the expression of the *UCP1* gene in SAT [105]. IS increased by 28.0% compared to the baseline in the group receiving 25 mg of OCA (*p* = 0.019) and by 20.1% in the group receiving 50 mg of OCA (*p* = 0.060), while it decreased by 5.5% in the placebo group. A significant reduction in BMI was observed only in the group receiving 50 mg of OCA. In the groups receiving OCA, there were also increases in serum levels of LDL cholesterol and FGF19 (doubling in the 35 mg of OCA group and tripling in the 50 mg of OCA group), associated with a reduction in levels of 7alpha-hydroxy-4-cholesten-3-one and endogenous bile acids, indicating FXR activation [105]. Thus, there is indirect evidence suggesting that FXR agonists can enhance browning in WAT.

### 3.13. Glucagon-like Peptide-1 Receptor Agonist

GLP-1RA are registered for the treatment of obesity and T2DM and, to date, have shown to be among the most effective drugs for reducing BM and improving MH.

#### 3.13.1. Experimental Studies

In a study conducted by Beiroa et al., higher expression levels of thermogenic genes such as *Cidea*, *Ucp1*, *Ucp3*, and *Prdm16* and batokines (FGF21, BMP7) were found in the BAT of mice treated with liraglutide. Liraglutide also induced browning of the WAT of mice, achieving significant differences in the expression of both universal BAT genes (*Ucp1*, *Dio2*, *Prdm16*, *Cox8B*) and beige adipocyte marker genes (CD137) 24 h after the injection [88,106]. In a study conducted by Gutierrez et al., liraglutide was administered intraperitoneally to mice, demonstrating an increase in blood IL-6 levels and an increase in IL-6 receptor signaling in the AT. These effects were accompanied by WAT browning and the increased expression of thermogenic genes. Both IL-6 blockade by antibodies and IL-6 receptor blockade abrogated the effects of liraglutide on browning [107]. Liraglutide in the experiment increased the activity of DIO2 in BAT, which suggests, among the mechanisms of its action on BAT activity, an increasing intracellular activation of TH [108,109]. Other works demonstrated that a significant part of the effects is realized by the activation of the GLP1 receptor in the brain. When liraglutide was administered centrally to mice, it stimulated BAT thermogenesis, the browning of WAT and decreased food intake with subsequent weight loss. It is interesting to note that GLP-1RA exhibits properties akin to sirtuin mimetics and PPAR alpha agonists to a certain extent. In an experimental study utilizing exenatide, increased levels of lipolytic signaling proteins, including SIRT1, were observed. The levels of RCTS alpha, PGC1 alpha, UCP1, and SIRT1 proteins significantly increased following exenatide treatment [110].

Another discussed mechanism of action is the modulation of the microbiota composition [111], which facilitates the increased production of short-chain FAs (such as butyrate and lactate) that are also considered stimulators of browning [3].

#### 3.13.2. Clinical Trials

The results of clinical trials in humans are ambiguous. Clinical trials demonstrated a decrease in energy expenditure according to indirect calorimetry in patients with T2DM with obesity during treatment with liraglutide for 26 weeks. At the same time, the volume of BAT measured by MRI in the supraclavicular depot did not change. In our study, liraglutide therapy for 6 months in obese patients without impaired glucose metabolism resulted in an increase in metabolic BAT volume as measured by PET/CT with 18F-FDG [8]. Also, clinical trial published by Janssen et al. demonstrated that GLP-1RA exenatide increased metabolic volume (+28%, *p* < 0.05) and glucose uptake (SUV- mean) (+11%, *p* < 0.05) of cervical and supraclavicular BAT depots in healthy young men [111]. Similar results were observed with the additional inclusion of the upper mediastinal, axillary, and paravertebral BAT depots—location zones of cBAT. At the same time, exenatide did not affect 18F-FDG uptake in subcutaneous or visceral depots, which also indicates that exenatide has an effect specifically on cBAT. Thus, the available studies have confirmed the ability of GLP-1RA to activate cBAT in humans. The discordant results in the van Eyk et al. study may be due to the fact that patients with DM were included [112]. However, we did not find any clinical studies demonstrating the browning of WAT in the available literature.

More recently, 6-month semaglutide therapy in patients with type 2 DM and obesity was reported to be able to enhance adipose-derived stem cell (ADSC) proliferation and the adipogenesis of both white and beige adipocytes, demonstrating the restoration of ADSC renewal functions [113].

For dual and triple agonists involving combinations of incretins (GLP-1RA and the gastric inhibitory polypeptide receptor agonist) and incretins with glucagon (GLP-1RA and the glucagon receptor agonist), studies of effects on BAT are just beginning. In an experimental study including male C57BL/6 mice, the effects of a high-fat diet and cotadutide (GLP-1RA and glucagon) therapy on BAT were examined. A high-fat diet induced a whitening of AT with the reversal of BAT development by cotadutide therapy. Cotadutide increased the expression of thermogenesis marker genes (*Ppara*, *Ucp1*), beta3AR, IL-6, decreased pro-inflammatory markers and improved angiogenesis. Thus, experimental data demonstrated the thermogenic effect of cotadutide on inducible rBAT in obese mice, labeling this effect as one of its mechanisms of action [114].

### 3.14. Miglitol

In the time period of 2013–2015, two research groups from Japan in experimental studies on mice demonstrated systemic effects of miglitol, an inhibitor of alpha-glycosidases adsorbed from the GI tract, on the activity of BAT. These effects were realized through the BA (bile acid)-TGR5-cAMP-DIO2 pathway, which induces *UCP1* expression in BAT [115]. The results of the second research group showed that miglitol directly enhances beta3-adrenergic signaling by increasing *UCP1* mRNA and protein expression and the expression of four proteins involved in the signaling cascade (PKA, HSL, P38alphaMAPK and PGC1 alpha) in mice with obesity induced by high-fat diet [116]. Meanwhile, in the subsequent period, we did not find studies that confirmed or disproved these results in humans (Appendix A).

## 4. Conclusions

Based on the understanding formed in recent years, BAT is present in adults, although its activity is variable and depends on age, sex, BMI, and a number of other parameters. Its quantitative assessment is severely hampered by the huge number of factors affecting its activity and by the fact that current non-invasive diagnostic methods allow us to assess only active BAT, while accurate assessment of the volume of inactive BAT is challenging [16]. In any case, the cBAT represents only about 2% of the adult AT volume, and its contribution to MH is unlikely to be leading [6]. Moreover, its amount does not change significantly during adult life, although its activity increases under the influence of thermogenic inducers. A much greater role may belong to rBAT, which is highly inducible. Beige adipocytes can be generated from precursor cells by differentiation, from mature white adipocytes by transdifferentiation, or activated from “dormant” beige adipocytes through the activation of various molecular genetic mechanisms [117,118]. The quantitative limit of rBAT has not been determined, but it is unequivocally much larger than the amount of cBAT in the body, and, therefore, its potential as a target for interventions is more extensive. The range of possibilities for beige AT to activate thermogenesis is also broader and is realized through both UCP1-dependent and UCP1-independent (creatine cycle, Ca++ cycle mechanisms). In addition, brown and beige AT differ in localization. According to recent views, beige adipocyte precursor cells and beige adipocytes are located in the perivascular, epicardial and retroperitoneal zones [119]. The de-browning of AT of these localizations is associated with an increased risk of vascular and renal pathology, respectively. This leads one to hypothesize that beige AT may play a major role in vascular, cardiac, and renal health by participating in preventing the development and progression of atherosclerosis, heart failure, and chronic kidney disease [120].

Meanwhile, a number of questions on these two types of thermogenic AT (brown and beige) remains a matter of debate. In particular, there is an opinion that, in adult humans, the classic BAT disappears, and all BAT is represented by beige AT, which makes its study even more topical. In a pathologic study conducted by Sharp LZ et al. on the AT of various localizations in humans, the universal BAT markers *UCP1*, *PRDM16*, *CIDEA*, PGC1a and the beige AT markers *TMEM26*, *CD137*, *TBX1* were expressed in almost all localizations of BAT (supraclavicular, retroperitoneal, mesenteric, retrosternal). No expression of markers of either cBAT or beige AT was detected in white WAT. However, when evaluating these results, especially in the WAT, it should be kept in mind that this was a postmortem tissue evaluation, which could affect the result. Markers of classic BAT (*ZIC1*) were not expressed in any of the AT sites [121]. And if we take the position of these authors, then all the therapeutic agents discussed in this review are inducers of beige AT. Meanwhile, in vivo studies evaluating SAT biopsies have demonstrated the expression in SAT from adult patients with the DM of both markers of beige AT—*TMEM26* and markers of classic brown AT (*ZIC1*) [122], which leaves the issue of identifying selective inducers relevant.

There are also contradictions regarding the attribution of individual markers to a certain type of BAT. For example, in some publications, *LHX8* is described as a marker of classic BAT [121], in others, as a universal marker of BAT [12], *Epsti1* is labeled as a marker of classic BAT in some studies [121], in others as a selective marker of beige AT [12], and, in others, as a universal marker of BAT [3]. The accurate mapping of beige and classic brown AT genes is essential for the identification of selective inducers.

In turn, beige AT is thought to be inducible and at least enters a dormant state under thermoneutral conditions. However, under experimental conditions, many selective markers of beige AT are highly expressed under thermoneutral conditions. Whether or not these markers will be detectable in humans under thermoneutral conditions and whether or not their expression is associated with metabolic health parameters is a question for future research.

Therefore, as a target for therapy, beige AT appears to be a more important target. However, our attempt to separate cBAT and rBAT inducers was not very successful, indicating that research in this area is just beginning. Many of the mechanisms are indeed similar. For example, exposure to cold has been found to similarly activate brown and beige adipocytes in adult humans, resulting in increased energy expenditure. The regulation of glucose and lipid metabolism by cBAT and rBAT also occurs in a similar manner. Meanwhile, analyzing approaches to assess the effects of different factors and drugs on specific mechanisms is just beginning. It is necessary to understand how long the pool of beige adipocytes is maintained after exposure to an inducer and the relationship between the duration and intensity of the inducing stimulus and the duration of maintenance of the beige adipocyte pool. Since the universal effect of drugs that activate the expression of BAT marker genes is to improve MH parameters, such as IS, obesity-associated lipids (TGs) and glucose levels [7,117,123] and the inhibition of athero- and carcinogenesis [3], this task seems important. Reduction in BM is not a key outcome of the activation of cBAT or browning [6]. Moreover, the dynamics of BM with these agents is highly variable, due to their effects on the activity of other tissues [15]. For example, the activation of PPAR gamma is accompanied by increased adipogenic differentiation in both WATe and BAT, and the resulting effect of PPAR gamma agonists is an increase in BM, not a decrease [15]. An equally important point is the systemic enhancement of sympathoadrenal effects when targeting the key mechanism of induction of both BAT activity and browning—activation of the betaAR. Selective action on β3AR reduces but does not eliminate this problem. Given the likely much greater contribution of rBAT formation to improving MH, the use of interventions that prioritize its formation seems an attractive idea. However, the available literature is extremely poor in providing data on exposure selectivity, and most studies performed using AT biopsies have examined the expression of universal marker genes in both cBAT and rBAT. Only a few studies have shown the increased expression of selective markers of rBAT (*TMEM26*, *CD137*) in response to physical activity, metformin, and resveratrol, and in experiments, for GLP1RA. These drugs, which we designated as sirtuin mimetics in this review, have shown the best safety profile of all the drugs discussed. This raises the question of whether it is worthwhile to further develop studies regarding the evaluation of sirtuin mimetics as inducers of rBAT. Another feature of these drugs is the widest spectrum of beneficial effects—not only browning activation, but also anti-aging, anti-tumor effects. This makes them an even more attractive choice in therapy.

Knowledge of the molecular and genetic pathways of beige AT activation, which can be maintained for a long time, may help both in optimizing the use of old drugs and in the development of new ones. For example, the ability of PPAR gamma to stimulate the differentiation of preadipocytes into white or brown/beige adipocytes appears to be determined by its associated coactivators. Again drawing attention to sirtuin mimetics, we should note that it is SIRT1 that deacetylates PPAR gamma, enabling its binding to PRDM16 in brown/beige adipocytes [124]. In contrast, binding to TLE3 leads to the suppression of PPAR gamma expression of BAT genes and the stimulation of white adipocyte genes [125], marking the prospect of combining PPARgamma agonists with SIRT1-mimetics. MicroRNA-based therapies may emerge as novel targeting approaches in therapy. For example, miR-196a activates C/EBPbeta by suppressing *HOXC8* expression, selectively increasing beige adipogenesis [126]. miR-155 suppresses C/EBPbeta expression, and its ablation increases both brown and beige adipogenesis in mice [127].

At the same time, the weaker effect of sirtuin mimetics such as metformin, when administered systemically, compared to other drugs, makes us pay close attention to the development of methods of the targeted delivery of these drugs to SAT. Targeted delivery may also become a relevant option for agents that, when systemically administered, have significant negative effects, such as tachycardia and increased BP with β3ARa. Further research in this direction will help us in solving the problem of increasing energy expenditure in the body as another direction in the treatment of obesity and metabolic syndrome.

## Figures and Tables

**Figure 1 ijms-25-13295-f001:**
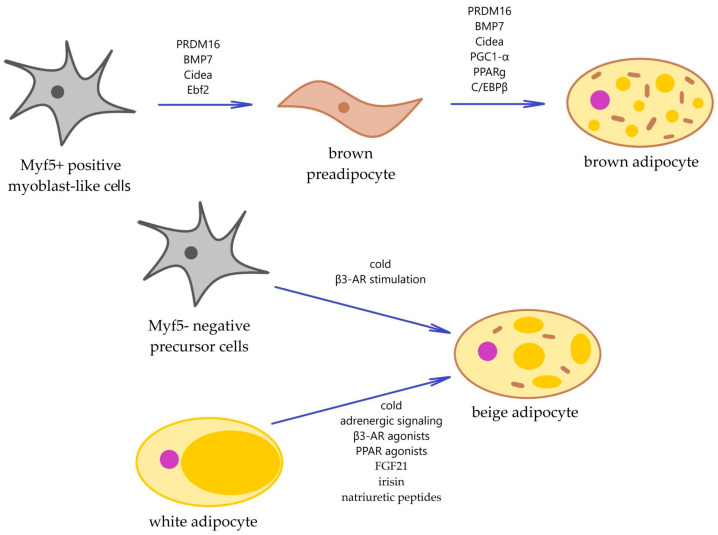
Origin of brown and beige adipocytes. β3-AR–beta3-adrenoreceptors; DIO2-type 2 deiodinase; PPAR-peroxisome proliferator-activated receptor; FGF21-fibroblast growth factor 21.

**Table 1 ijms-25-13295-t001:** Genetic markers, unique for various adipose tissue types (adapted and added from Ghesmati Z et al. [3]).

Parameters	BAT	cBAT	rBAT
Specific expression of genes	*UCP1* *PRDM16* *PPARGC1A* *ELOVL3* *CIDEA* *LHX8* *PPARG/A* *EPSTIi1* *DIO2* *COX8b* *SIRT1*	*DMP7*, *EBF2 and 3*, *EVA1*, *MYF5*, *PDK4*, *PREX1*, *ZIC1*, *HSPB7*, *miR-206*, *miR-133b*, *OPLAH*, *ACOT2*, *FBXO31*, *PPARGC1A*	
Precursor cells		Myf5+ positive myoblast-like cells differentiate into brown preadipocytes under the influence of PRDM16, BMP7, CIDEA, and Ebf2, and into brown adipocytes under the action of transcriptional regulators PRDM16, BMP7, CIDEA, PGC1-α, PPARg and C/EBPβ.	Myf5- negative precursor cells, under cold and β3-AR stimulation, can directly differentiate into beige adipocytes, or under the influence of cold exposure, adrenergic signaling, β3-AR agonists, PPAR agonists, FGF21, irisin, and natriuretic peptides, transdifferentiate from white adipocytes.
Activating microRNAs	MiR-30, MiR-32, MiR-455	MiR-365, MiR-193b, MiR-182, MiR-203, MiR-328, MiR-129, MiR-378	MiR-196a, MiR-let-7, MiR-26.
Inhibiting microRNAs	MiR-34a, MiR-155, MiR-133, MiR-27	MiR-106b, MiR-93	MiR-378, MiR-125
The main thermogene	UCP1-dependent		UCP1-independent—via sarco/endoplasmic reticulum Ca^2+^-ATPase 2b (SERCA2b) and ryanodine receptor 2, creatine cycle

BAT—brown adipose tissue; cBAT—classic brown adipose tissue; rBAT—recruitable brown adipose tissue.

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
