# Peer review of "Brown and Beige Adipose Tissue: One or Different Targets for Treatment of Obesity and Obesity-Related Metabolic Disorders?"

_ijms, 2024, doi:10.3390/ijms252413295_

Round 1
Reviewer 1 Report
Comments and Suggestions for Authors
Regarding the title, I did not understand the meanings of “obesity-determenated metabolic disorders”. Moreover, I did not understand what the term “determenated” indicated. This point on the title is fatal error.
I am afraid that the expression of “Bathokine” is incorrect. Many published papers have used the terms BATokine or Batokine in my awareness.
Introduction is too much. The author must summarize physiological functions of brown adipose tissue (BAT). Formerly, the detection of cold-circumstances induced BAT by FDG-PET was famous and interesting, but BAT activation by chemical and pharmacological agents is the main theme in this paper. We, the readers of IJMS, have no need to precisely know the relationship between FDG-PET and BAT. The topic of FDG-PET CT and BAT is a past thing among the researchers of BAT.
I admit the authors have abundant knowledge on basic and clinical research on BAT to develop and establish BAT activation by the drugs. However, they should be aware of the research on BAT activation by miglitol, erythropoietin and Ang1-7.
The author must write the manuscript using tables and figures so that we can easily understand their thinking.
Author Response
Dear reviewer!
:
The authors sincerely thank you for your thorough and insightful analysis of our work. Regarding your comments:
Regarding the title, I did not understand the meanings of “obesity-determenated metabolic disorders”. Moreover, I did not understand what the term “determenated” indicated. This point on the title is fatal error.
You are absolutely right, the word was not correct. We have replaced it in the title with "obesity-related metabolic disorders."
I am afraid that the expression of “Bathokine” is incorrect. Many published papers have used the terms BATokine or Batokine in my awareness.
Yes, you are absolutely right, thank you very much for your comment. The term “Bathokine” has been corrected to Batokine throughout the text.
Introduction is too much. The author must summarize physiological functions of brown adipose tissue (BAT). Formerly, the detection of cold-circumstances induced BAT by FDG-PET was famous and interesting, but BAT activation by chemical and pharmacological agents is the main theme in this paper. We, the readers of IJMS, have no need to precisely know the relationship between FDG-PET and BAT. The topic of FDG-PET CT and BAT is a past thing among the researchers of BAT.
We have reviewed the Introduction section and completely agree with you. It was excessively lengthy. We have shortened this section and allocated a new section titled "Modern Understandings of the Role of Adipose Tissue in Metabolic Health" for discussion general aspects. Additionally, we have revised the information regarding the role of FDG-PET CT in the indication of BAT and we shortened this part. At the same time, we can't avoid discussing it, because most studies use this technique to dynamically assess the effect of drugs on BAT.
I admit the authors have abundant knowledge on basic and clinical research on BAT to develop and establish BAT activation by the drugs. However, they should be aware of the research on BAT activation by miglitol, erythropoietin and Ang1-7.
In our review, we tried to focus on those drugs that have been evaluated not only in experimental studies but also in clinical trials. Therefore, the drugs you mentioned and a number of other drugs are not discussed in this review. However, we felt it necessary to follow your recommendation and have added information about these medications.
The author must write the manuscript using tables and figures so that we can easily understand their thinking.
Yes, you are absolutely right that figures and diagrams make it easier to perceive information. However, the journal has limitations regarding the number of figures and tables. The review already contains two large tables. We have introduced additional figures within the permitted limits.
Once again, we would like to thank you for your important and useful recommendations for improving our article, and we sincerely hope that we were able to improve it sufficiently.
Сorrections in accordance with your remarks are highlighted in yellow.
Sincerely,
The Authors
Reviewer 2 Report
Comments and Suggestions for Authors
Mayor revision
1- The absence of numbering in the sentences or paragraphs makes it impossible to correctly identify errors, including typographical ones.
2- The introduction of the manuscript sets a detailed and technical overview of adipose tissue, specifically focusing on the different types of adipose tissue, their functions, and the mechanisms involved in their activation and differentiation. However, it would be necessary to consider the use of headings and subheadings to delineate the different sections. Moreover, the literature provides strong support for the discussion of the different mechanisms involved in beige adipocyte formation. However, including opposing viewpoints or recent research could offer a more balanced perspective.
3- Review bibliography and ensure that all references are up-to-date and relevant to maintain credibility.
4.- Certain sentences are excessively lengthy and incorporate multiple clauses, which may impede comprehension. For example (one of many, but I can't point them out because there are no identifying numbers): the statement regarding the synergistic action of TH and SNS should be simplified to enhance clarity and readability.
5.- The flow of the full text could be improved by clearly delineating sections for mechanism of action, experimental findings, clinical implications, and limitations. Please include those sections in all sections in the same way, uniformly. This would facilitate better comprehension and allow readers to follow the argument more easily.
6.- The conclusion summarizes the main points discussed in the review. However, it would benefit from a more organized structure. Consider dividing it into separate paragraphs that address different themes, such as the significance of cBAT vs rBAT, implications for treatment strategies, and directions for future research.
7.- It would be necessary to make a more detailed and organized graphical abstract, providing more comprehensive information and presenting the information in a clearer, more structured manner.
Minor comments
1- Please, correct the term bathokine, and replace it with 'batokine.'
2- The authors should use the appropriated nomenclature for human genes throughout the manuscript. Please, write the gene symbols in italics, all uppercase, without Greek letters and without hyphens.
3.- The authors must place a space after dots, commas, and before and after parenthesis throughout the entire text.
Comments on the Quality of English Language
The English of the manuscript needs a deep revision.
Author Response
Dear reviewer!
The authors sincerely thank you for your thorough and insightful analysis of our work. Regarding your comments:
We fully agree with you that using headings and subheadings to mark different sections will improve the perception of the text. We have introduced a more detailed rubrication of the text. Thank you for your useful recommendation!
Moreover, the literature provides strong support for the discussion of the different mechanisms involved in beige adipocyte formation. However, including opposing viewpoints or recent research could offer a more balanced perspective.
Thank you for your valuable advice, we tried to do this in the Conclusion.
We also checked the list of sources and tried to edit the text, removing excessively long sentences. In the conclusion, we divided the text into thematic paragraphs. We supplemented and edited the graphic abstract.
We would like to thank you for your important and useful recommendations for improving our article, and we sincerely hope that we were able to improve it sufficiently.
Сorrections in accordance with your remarks are highlighted in blue.
Sincerely,
The Authors
Round 2
Reviewer 2 Report
Comments and Suggestions for Authors
The authors should use the appropriated nomenclature for genes throughout the manuscript (PPARG, PGC1A, FGF21, BMP7, etc). Please, write the gene symbols in italics, all uppercase, without Greek letters and without hyphens in the case of human genes, or italized, first letter uppercase, rest lowercase in tha case of rat genes, as it corresponds according with the references!!!
Lines 477-482: correct the format
Lins 610-675: correct the format
Author Response
- The authors should use the appropriated nomenclature for genes throughout the manuscript (PPARG, PGC1A, FGF21, BMP7, etc). Please, write the gene symbols in italics, all uppercase, without Greek letters and without hyphens in the case of human genes, or italized, first letter uppercase, rest lowercase in tha case of rat genes, as it corresponds according with the references
Corrections have been made, highlighted in red
-
Lines 477-482: correct the format
Lins 610-675: correct the format
Lines 477-482 and 610-675 - format corrected